# A Preliminary Study on Cross-Reactivity of Heat-Treated Quail and Hen’s Egg White Proteins in Young Children

**DOI:** 10.3390/nu13072172

**Published:** 2021-06-24

**Authors:** Jeongmin Lee, Purevsan Gantulga, Changhoon Lee, Kyunguk Jeong, Eunjoo Lee, Sooyoung Lee

**Affiliations:** 1Department of Pediatrics, Yonsei University Wonju College of Medicine, Wonju 26426, Korea; ejeongmin@yonsei.ac.kr (J.L.); 1111hoon@gmail.com (C.L.); 2Department of Pediatrics, Ajou University School of Medicine, Suwon 16499, Korea; gpsn0321@yahoo.com (P.G.); k.u.jeong@gmail.com (K.J.); 3Department of Pediatrics, Inje University Haeundae Paik Hospital, Busan 48108, Korea; pro1595@gmail.com

**Keywords:** hen’s egg allergy, quail egg, cross-reactivity

## Abstract

We investigated the effects of different types of heat treatments on hen’s egg white (HEw) and quail egg white (QEw) proteins and their cross-reactivity in young children. Crude extracts of raw and water-boiled HEw and QEw and commercially developed stone-baked HEw were prepared. Sodium dodecyl sulfate–polyacrylamide gel electrophoresis (SDS-PAGE) was then performed. Patients diagnosed with HEw allergy were enrolled, and pooled sera were tested with each extract in an enzyme-linked immunosorbent assay (ELISA)-inhibition test. A skin prick test (SPT) and oral food challenge (OFC) were also performed. The median age of 12 patients was 2.5 years. SDS-PAGE results revealed strongly stained bands for the ovomucoid of boiled HEw and QEw, while stone-baked HEw displayed remarkable changes for all protein fractions. In the ELISA-inhibition test, pre-incubation of the sera led to a profound decrease, moderate decrease, and minimal decrease in the amount of IgE binding to boiled and raw HEw, boiled and raw QEw, and stone-baked HEw proteins, respectively. SPTs and OFC demonstrated cross-reactivity values of 41.7% (5/12) and 16.7% (2/12) for boiled QEw and stone-baked HEw, respectively. We observed moderate cross-reactivity between QEw and HEw. Boiling had a limited effect on altering egg allergenicity. Commercially developed, stone-baked HEw can be an alternative food for children with HE allergy.

## 1. Introduction

Avoiding problematic allergens is an essential part of the management of food allergies. Moreover, patients with hen’s egg white (HEw) allergy are recommended to limit the consumption of all poultry eggs owing to the possibility of cross-reactivity [1,2]. In the Korean food allergen labeling system, quail eggs (QEs) are labeled as bird eggs [3]. In the medical literature, there are reports of cross-reactivity to duck or goose eggs in patients with HEw allergy worldwide. However, it is not clear whether cross-reactivity to quail egg white (QEw) occurs in patients with HEw allergy [1,2,4]. A previous study reported that the co-sensitization rate in patients with HEw allergy, which was assessed through skin prick tests (SPTs) using raw QEw, was 59.2% [1]. The aforementioned study excluded patients with anaphylactic reaction to HEw, and the criteria for SPT readings were as follows: wheal diameter >5 mm for patients aged ≤2 years and >7 mm for patients aged >2 years. Another previous study [4] on the characteristics of allergens in QEw in five HE-tolerant adults diagnosed with QE allergy reported that the sera of most patients cross-reacted to the protein present in the 42 kDa band (ovalbumin), whereas the sera of a few patients reacted to the 97 kDa (ovotransferrin) and 35 kDa (ovomucoid) band proteins, indicating a high degree of homology with HEw [5]. 

Conversely, HEw proteins undergo changes in their antigenicity when exposed to heat [6]. A previous study on the changes in HEw proteins after heat treatment, wherein eggs were boiled in water at 90 °C for 10 min, reported that the protein band corresponding to ovalbumin weakened progressively, whereas that corresponding to ovomucoid remained stable. In general, it is difficult to conclude whether the cross-antigenicity between the proteins is solely responsible for the clinical symptoms, especially considering the variation in the antigenicity of QEw and HEw, which is dependent on the thermal treatment, despite the high degree of homology and the corresponding co-sensitization rate [7,8,9]. In the present study, we assessed the cross-antigenicity and reactivity of heat-treated QEw in a group of pediatric patients with HEw allergy, particularly in patients with a history of anaphylaxis. Additionally, we examined the antigenicity of stone-baked HEw, which is commercially available in South Korea. Commercially developed stone-baked HEs are processed on elvan stones that are heated under far-infrared light without the use of water. The process involves a gradual increase in temperature from 45 °C to 110 °C over a period of 24 h [10]. There is no known change in the antigenicity of egg whites (EWs) cooked using this method.

In the present study, we assessed hypersensitivity in patients with HEw allergy, including those showing an anaphylactic reaction to HEw and commercially developed stone-baked HEw. We also evaluated the cross-reactivity to heat-treated QEw, which was cooked using the home-cooking process, in patients with HEw allergy. The present study aimed to provide alternative dietary choices and contribute to the improvement of the quality of life of patients with HE allergy.

## 2. Materials and Methods

### 2.1. Patients and Sera

This study prospectively enrolled 12 patients who were diagnosed with HEw allergy by pediatric allergists at Yonsei Wonju Severance Christian Hospital between January 2020 and June 2020. Clinical history of any allergic reactions to well-boiled HEw and any form of QE was recorded. HEw-sIgE concentration was measured using ImmunoCAP (Thermo Fisher Scientific, Uppsala, Sweden). The inclusion criteria were as follows: (1) age ≤ 12 years; (2) HEw-sIgE concentration ≥ 0.35 kU/L; and (3) at least one confirmed occurrence of the following in the past 6 months—(a) HEw-induced anaphylaxis, (b) repeated (twice or more) allergic reactions to HEw, or (c) positive results on oral food challenge (OFC) using water-boiled HEw. Informed consent was obtained from the parents of seven patients for using the blood samples, especially serum. These blood samples were collected from the patients at the time of diagnosis and were stored at −40 °C until use.

### 2.2. Preparation of Extracts

Medium-sized HEs and QEs were purchased from a local store and boiled for 15 min in water (e.g., 100 °C). Stone-baked HEs were purchased from a local Korean market. The manufacturing process of quartz porphyry hen eggs was registered as a patent (KR0166422B1) in South Korea in 1998 [10]. The cuticle layer on the surface of the shell was washed and removed, and the temperature was increased through a supply of heated air and by filling the interior of commercial ovens used for this process with quartz porphyry. Although the method followed by different manufacturers may vary, the temperature is typically raised from a low temperature to approximately 120 °C over a period of 10 h. Subsequently, the high temperature was maintained for another 10 h. The application of heat to quartz porphyry results in the emission of far-infrared light, which prevents shell breakage and enables long-term preservation as the moisture is removed and the yolk is cooked evenly by appropriate heat transfer and thermal efficiency. Extracts of raw, boiled, and stone-baked HEw and raw and boiled QEw were prepared according to the following method: the samples were added to phosphate-buffered saline (pH 7.4; 1:1 *w*/*v*) and placed under stirring at 4 °C for 7 days. The extracts were centrifuged at 10,000 rpm for 1 h. The resulting supernatant was dialyzed in deionized water for 48 h (pore size cutoff: 3.5 kDa). Subsequently, the samples were freeze-dried at −70 °C, and protein concentrations were measured according to the Bradford assay (Bio-Rad, Hercules, CA, USA) using a microplate reader.

### 2.3. Identification of Proteins in Each Extract by Sodium Dodecyl Sulfate–Polyacrylamide Gel Electrophoresis

The samples were analyzed by sodium dodecyl sulfate (SDS)–polyacrylamide gel electrophoresis (PAGE) according to the method described by Laemmli [11]. For SDS-PAGE, the processed extracts were prepared at a concentration of 5 µg/mL and mixed with loading buffer (0.5 M Tris-HCl, pH 6.8, glycerol, 10% SDS, 0.5% bromophenol blue, and 2.5% β-mercaptoethanol). The mixture was heated at 100 °C for 10 min and electrophoresed in 4–20% Tris-glycine gradient gels (Invitrogen, San Diego, CA, USA) at 120 V for 2 h, together with a marker (SeeBlue^®^plus2, Invitrogen). The marker was also electrophoresed according to the same method to analyze the results.

### 2.4. Enzyme-Linked Immunosorbent Assay-Inhibition Test

Enzyme-linked immunosorbent assay (ELISA)-inhibition tests for reactivity to boiled HEw were performed using pooled sera. Extracts of stone-baked and raw HEw and boiled and raw QEw were used as inhibitors, and each inhibitor was diluted to concentrations of 0.1, 1, 10, and 100 µg/mL. The inhibitors and pooled sera were allowed to react at room temperature for 2 h. The degree of IgE inhibition for boiled EW was calculated using the following equation: %inhibition = [(inhibited OD − uninhibited OD)/uninhibited OD] × 100, where OD is the optical density.

### 2.5. Skin Prick Test and Oral Food Challenge

SPTs were performed against raw, boiled, and stone-baked HEw and boiled QEw extracts. All the extracts were used at a concentration of 0.1 mg/mL. All SPTs were performed on the volar side of the lower arm of the children. Histamine (Lofarma Allergeni, Milan, Italy) and normal saline were used as positive and negative controls, respectively. The mean values of the longest and the midpoint orthogonal diameters (mm) of the wheals were measured after 15 min to determine the average diameter. Patients were excluded for OFCs when (a) An SPT result was considered to be positive if the mean diameter of the test wheal was equivalent to or greater than that of the positive control wheal. (b) Patients with a mean wheal diameter smaller than that of the positive control but greater than that for boiled HEw were also considered to be positive.

Patients whose SPT results were below the cutoff value and those whose parents provided informed consent underwent the OFC. Regardless of the SPT results, patients with a history of anaphylaxis to HEw were excluded from undergoing the OFC test using boiled HEw. Each OFC consisted of several escalating doses of proteins. Typically, HEw OFC commenced with a small dose of the HEw protein, that is, 370 mg (one-eighth of one medium-sized HEw). Subsequently, the dose was increased to 750 mg (one-fourth of one medium-sized HEw), 1.5 g (half of one medium-sized HEw), and 3 g (one medium-sized HEw). Each dose was administered orally at 20-min intervals. If the patients were underweight (≤15 kg), then the OFC was initiated with a smaller dose of the HEw protein (180 mg) to reach a final dose of 1.5 g. For patients with a history of anaphylaxis, the starting dose was reduced by half (Appendix A). The protocol for OFC using QEw is shown in Appendix A. The R program (version 3.0.2; R Foundation for Statistical Computing, Vienna, Austria) was used for performing statistical analyses. Normality was tested, and *p*-values for continuous variables were calculated using the Wilcoxon rank-sum test. Fisher’s exact test was used to compare categorical variables. The significance level was set at *p* < 0.05.

## 3. Results

### 3.1. Clinical and Immunological Characteristics of Patients

The clinical and immunological characteristics of the 12 patients are shown in Table 1. The median patient age was 2.5 years (interquartile range (IQR), 1.0–6.5 years), and the study sample comprised eight male and four female patients. Of them, seven (58.3%) had a history of immediate (less than 2 h) anaphylaxis after the consumption of boiled HEw, and five (41.7%) had a history of at least two incidents of acute urticaria. The patients were divided into two groups on the basis of their history of allergic reactions; the HEw-sIgE concentrations were significantly higher in the anaphylaxis group (26.3 kU/L; IQR, 19–34.7 kU/L) than in the urticarial group (4.3 kU/L; IQR, 3.8–4.5 kU/L) (*p* < 0.05). The most frequently observed comorbidity was atopic dermatitis (*n =* 11), followed by food allergies other than HEw allergy (*n =* 10), allergic rhinitis (*n =* 4), and bronchial asthma (*n =* 3). The most commonly identified comorbid food allergies were those to cow’s milk (*n =* 6) and wheat (*n =* 4). None of the patients had previously consumed the QEw.

### 3.2. Identification of the Proteins in Processed HEw and QEw Extracts

In SDS-PAGE, the separation patterns of boiled QEw and HEw showed notable changes in the protein fractions after 15 min of boiling. The protein bands from raw HEw were identified as 14, 28, 34, 40, 52, and 69 kDa. The 52 and 69-kDa bands were stable after boiling in water. The protein bands from raw QEw were identified at 14, 18, 28, 40, and 52-kDa. The 28-kDa protein band was enhanced in the boiled extract, whereas the 52-kDa protein band was slightly diminished. Protein bands were interpreted based on the findings of a previous report [5]. The presence of strongly stained bands for ovomucoid suggests that the antigenicity of ovomucoid of HEw and QEw remained stable during boiling. Conversely, the bands for ovalbumin became weakened by boiling. Transferrin and lysozyme were not detectable after 15 min of boiling. The commercially developed stone-baked HEw showed remarkable changes in all protein fractions. The bands for transferrin, lysozyme, ovalbumin, and ovomucoid, which remained stable despite being boiled, were barely detectable in the stone-baked HEw extracts (Figure 1).

### 3.3. In Vitro Analysis of the Cross-Reactivity between HEw and QEw

Moderate cross-reactivity between boiled and raw HEw and raw and boiled QEw was observed using competitive IgE ELISA-inhibition. Raw HEw showed a wide range of inhibition rates (inhibitory concentration (IC)50, 0.820 µg) in patients with HEw-sIgE ≥ 0.35 kU/L. The inhibition rates of boiled (IC20, 0.048 µg) and raw QEw (IC20, 0.283 µg) were in the medium range (approximately 30–50%), while those of the stone-baked HEw were in the low range (<30%) (Figure 2).

### 3.4. In Vivo Analysis of the Cross-Reactivity between HEw and QEw

SPT results for boiled HEw indicated a relatively high median value of 5.4 mm (IQR, 0.5–5.7 mm), compared to the positive control, which displayed a median value of 4.5 mm (IQR, 2.1–6.3 mm). Three who had a history of anaphylaxis proceeded with the OFC using boiled HEw because their parents wanted to know the symptom-eliciting dose. However, two developed anaphylaxis, and one developed cough. Furthermore, one among the four patients with SPT results for boiled HEw below the cutoff value (No. 8, SPT results: 4.8 mm, HEw-sIgE concentration: 4.98 kU/L) developed anaphylaxis. Alternatively, the lowest median SPT result among all the extracts was observed with stone-baked HEw, with a value of 2.2 mm (IQR, 0.0–4.2 mm). For baked HEw, the anaphylaxis group displayed a median SPT result of 3.7 mm (IQR, 2.2–5.3 mm), which was significantly higher than that of the urticarial group (0.0 mm; IQR, 0.0–0.0 mm) (*p* < 0.05). Of the three patients with SPT to stone-baked HEw results above the cutoff value, two participated in the OFC because their parents wanted to know their cross-reactivity, and one (No. 5. SPT results: 6.0 mm, HEw-sIgE concentration: 22.9 kU/) passed. The SPT and/or OFC results revealed that only 16.7% (2/12) of the patients demonstrated cross-reactivity to stone-baked HEw. The median SPT value for raw HEw was 3.5 mm (IQR, 4.0–4.6 mm), and 66.7% (8/12) of the patients displayed values above the cutoff. Although the median SPT result for boiled QEw was higher than that for stone-baked HEw (3.6 mm; IQR, 0.5–5.7 mm), it was lower than that for boiled HEw. Three patients showed median SPT results above the cutoff value for boiled QEw. Moreover, one of the nine patients with SPT results for boiled QEw below the cutoff value displayed a larger mean wheal diameter than that observed with boiled HEw. Therefore, the OFC for boiled QEw was offered to eight (66.7%) patients, among whom four underwent OFC with parental/guardian consent; of them, three patients passed the challenge. The SPT and/or OFC results showed cross-reactivity in 41.7% (5/12) of the patients. The patient (No. 12) who did not pass the OFC was a 1-year-old infant who developed rashes and subjective irritability. The mean wheal diameter with regard to the SPT for boiled QEw was 0 mm.

SPTs were performed to identify patients with results that were below the values for both the positive control and boiled HEw. Patients without a history of anaphylaxis underwent the OFC and had negative predicted outcomes. However, in situations involving positive predicted outcomes, only if the parents were willing to consent for an OFC, these OFCs were conducted, and a total of 18 OFCs were performed in this study. Among the 14 cases with negative predicted outcomes, two patients (No. 8: boiled HEw and No. 12: boiled QEw) became symptomatic, and three of the four patients with positive predicted outcomes were symptomatic. Hence, the positive predictive value was 68.8% (95% confidence interval (CI), 22.7–94.3), with a negative predictive value of 89.1% (95% CI, 73.4–96.0), sensitivity of 60.0% (95% CI, 14.7–94.7), specificity of 92.3% (95% CI, 64.0–99.8), and accuracy of 85.2% (95% CI, 60.8–97.2) (Table 1, Figure 3).

## 4. Discussion

Previous studies have reported that despite the variations in the antigenicity of allergens present in HEw according to the type of heat treatment, the majority of the patients with HEw allergy displayed tolerance toward HEw cooked at high temperatures [7,8,9]. Nevertheless, the aforementioned studies used a baked form of HEw treated with wheat and/or sugar powder. Consequently, they are not free from the matrix effect [6,12]. Additionally, there are challenges associated with home baking using a specialized form of mixed powder for patients with multiple food allergies, as they find it difficult to consume commercially available baked products. In the current study, 75.0% (9/12) of the patients were concomitantly allergic to cow’s milk or wheat. Furthermore, baking is not popular in South Korea, and the specifications for household ovens are variable. Hence, an alternative diet that involves home baking is impractical. Nevertheless, the ability to consume baked forms of HEw is useful when trying a low allergenic diet to achieve better quality of life and nutrition in patients with HEw-anaphylaxis, particularly compared to those who are unable to consume such products [7,8]. The present study aimed to examine whether the commercially developed, stone-baked HEw could be used as a hypoallergenic diet in HEw-anaphylaxis patients and to evaluate the cross-reactivity with QEw, which could be used as a potential alternative diet.

Although the criteria for SPT results used for the interpretation of sensitization vary in the literature [1,13,14], a mean wheal diameter greater than 3 mm in relation to that of the negative control is commonly interpreted as the occurrence of sensitization [15]. Nevertheless, in this study, it was presumed that the minimization of false-positive rates would be beneficial for a confirmative diagnosis of food allergy [16]. Hence, patients who presented with a mean wheal diameter equivalent to or greater than that of the positive control (class 3) were considered to be positive, and patients with a mean wheal diameter smaller than that of the positive control but greater than that for boiled HEw were also considered positive; these patients were technically not eligible for OFC but were offered the option of undergoing it. In principle, patients with a history of anaphylaxis were excluded from the OFC, regardless of SPT results. These factors are believed to have increased the quality of the diagnosis in the current study. Furthermore, our method has a higher accuracy than the methods used in previous studies that examined the specificity of SPT with HEw in patients of similar age groups [13,14]. Although the cutoff value of SPT results was used as a criterion for performing the OFC, this criterion may not be suitable for pediatric patients and should be modified depending on the age and the guardians’ preferences, which is reflective of real-world clinical practices. Patient No. 3 was a one-year-old patient whose guardian was passive toward the OFC. Conversely, the guardian of patient No. 1, who had multiple food allergies and anaphylaxis, wanted to know the possibility of using QEs or baked HEs as an alternative to boiled HEs and to learn about the appropriate dose for boiled HEw. Moreover, it is often difficult to perform OFCs in pediatric patients aged <2 years. When administering the OFC for QEw to patient No. 12, the typical hassles associated with such situations were encountered, wherein the infant refused the food or fell asleep while crying excessively, which made it difficult to repeat or increase the dose of the food challenge, even though the symptoms were subjective. School-aged patients often demonstrate high resistance to unfamiliar foods that they have not consumed for a long time, which makes it difficult or even impossible to perform an OFC, especially in a blinded manner. Among the four patients who refused the OFC for boiled QEw, patient No. 6 refused to consume the eggs after the OFC for stone-baked HEw. Such challenges reflect the circumstances in which the diagnosis and treatment of pediatric food allergies must be individually tailored. 

A review of the medical history of the patients revealed that none of them had previously consumed QEw. This is probably attributable to their adherence to the treatment guidelines for HEw allergy. A few previous studies on the cross-antigenicity between QEw and HEw have supported the broad avoidance of all bird egg types in the presence of HEw allergy. In particular, a previous study that investigated cross-antigenicity using SPTs demonstrated a co-sensitization rate of more than 50% [1]. However, the cutoff values for the SPT results adopted in studies and the reported false-positive rates differ, depending on patient age and study design. In addition, the types and methods of thermal treatment of extracts used for SPTs have been inconsistent among previous studies [8,13,14]. Furthermore, the HEw extracts used in previous studies for SPTs were raw or freeze-dried, unlike the HEw-containing products usually available in the market. In the present study, HEw and QEw extracts were boiled in a manner similar to that used for domestic cooking. The resultant protein bands in vitro were found to be weaker, and cross-reactivity in vivo was found to be lower (41.7%) than the results reported in previous studies [1,2]. Moreover, cross-reactivity in vivo was not correlated with the severity of previous allergic reactions to HEw or the concentration of HEw-sIgE. Two of the patients with anaphylaxis (28.6%) consumed boiled QEw as an alternative diet. Accordingly, there is a need to evaluate whether the diet restrictions imposed by the present guidelines are too broad and too uniform for patients. Cross-reactivity to stone-baked HEw was the lowest among all extracts in vitro and was reported to be only 16.7% in our in vivo study, which indicates its potential value as a hypoallergenic alternative in the majority of patients. Boiled HEw can be easily prepared at home, but the protein bands corresponding to the ovomucoid, which the majority of pediatric patients are sensitized to [17], remained stable for boiled HEw. This is concurrent with the results reported in previous studies that demonstrated a high density of the band corresponding to ovomucoid on SDS-PAGE [8,18]. Taking the aforementioned facts into consideration, we can conclude that the symptoms are inevitable when boiled eggs are consumed.

The present study had certain limitations. First, the potential alternative diets for HEw in patients with QEw allergy were not explored, owing to the rarity of QEw allergy in children. Furthermore, some young patients were not asked to undergo the OFC. Therefore, interpreting the cross-reactivity of QEw or baked HEw in patients diagnosed with HEw based on this preliminary finding, which has relatively weak statistical evidence, is insufficient for clinicians. Regardless of the limitations, based on the current findings, we strongly suggest tailored alternative diets to be actively recommended for patients with HEw allergy, even those with a history of anaphylaxis. Moreover, because our methodology showed a good diagnostic value for outcomes of OFCs, the outcomes of this study could help physicians elucidate alternative diets for patients diagnosed with food allergy. If active efforts are added for young patients, in the future, statistically stronger evidence could be gathered using more valuable data. Heat-treated QEw and stone-baked HEw used in the current study can be useful constituents of a hypoallergenic alternative diet, not only for providing nutritional benefits but also for developing immune tolerance. In addition to detailed clinical history taking and measuring HEw-sIgE concentrations, performing SPTs or OFCs for stone-baked HEw and boiled QEw to propose an alternative diet may be beneficial for the diagnosis and treatment of patients with HEw allergy.

## Figures and Tables

**Figure 1 nutrients-13-02172-f001:**
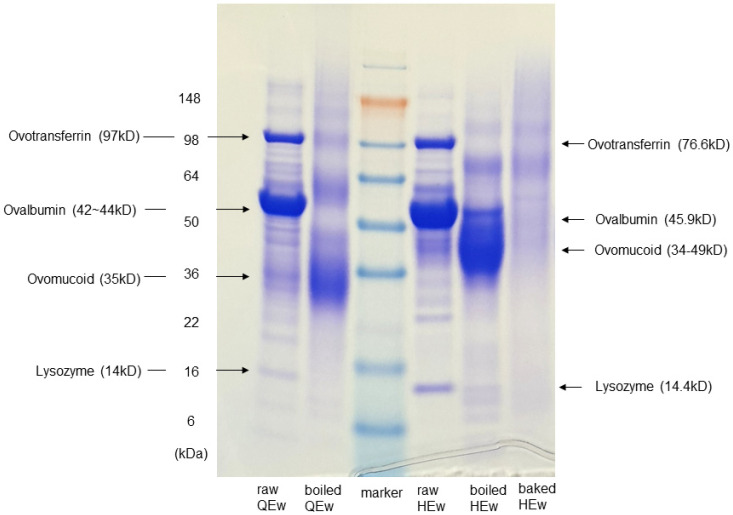
Sodium dodecyl sulfate–polyacrylamide gel electrophoresis (SDS-PAGE) analysis of the processed extracts of quail’s and hen’s egg whites. Concentration of extracts: 5 µg/mL. QEw, quail egg white; HEw, hen’s egg white.

**Figure 2 nutrients-13-02172-f002:**
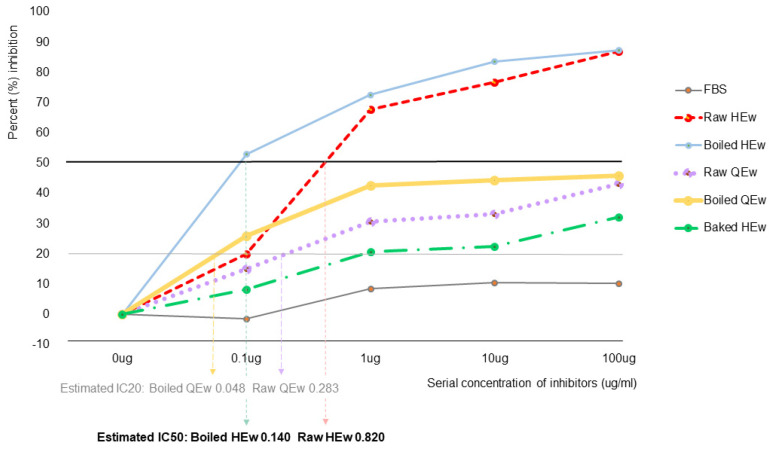
IgE-enzyme-linked immunosorbent assay-inhibition test with pooled sera using inhibitors. QEw, quail egg white; HEw, hen’s egg white.

**Figure 3 nutrients-13-02172-f003:**
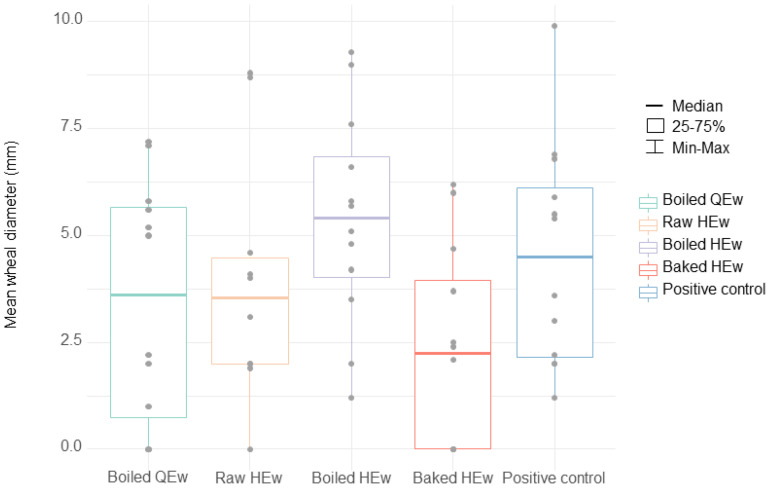
Boxplot showing the distribution of the results of the skin prick tests (mm, mean wheal diameter) for boiled quail egg white, raw hen’s egg white, boiled hen’s egg white, and stone-baked hen’s egg white extracts. The boxes show the interquartile ranges with the median value as the midline (*p =* 0.113). QEw, quail egg white; HEw, hen’s egg white.

**Table 1 nutrients-13-02172-t001:** Clinical and immunological characteristics of patients.

No.	Age (years)	Sex	Clinical History of Allergy		HEw-sIgE (kU/L)	Comorbidity		SPT (wheal diameter, mm)				OFC		
			To Well-Boiled Hen’s Egg	To Any Form of QE			Raw HEw	Boiled HEw	Stone-Baked HEw	Boiled QEw	Histamine	Stone-Baked HEw	Boiled HEw	Boiled QEw
1	3	M	Anaphylaxis	Unknown	97.3	AD, FA (perilla, walnuts, soybean), Asthma,	4.6	9.3	3.7	0.0	2.0	Fail: anaphylaxis	Fail: cough	Pass
2	3	M	Anaphylaxis	Unknown	38.2	AD, FA (perilla, wheat)	8.7	6.6	6.2	5.6	9.9	Pass	Excluded	Pass
3	1	F	Anaphylaxis	Unknown	31.1	AD, FA (wheat)	4.0	4.2	2.4	2.2	2.2	Excluded	Excluded	Excluded
4	8	M	Anaphylaxis	Unknown	26.3	AD, FA (CM, wheat), AR, Asthma	8.8	5.7	4.7	5.8	6.8	Pass	Excluded	Excluded
5	6	M	Anaphylaxis	Unknown	22.9	AD, AR, FA (CM)	3.1	7.6	6.0	7.2	5.9	Pass	Fail: anaphylaxis	Excluded
6	7	M	Anaphylaxis	Unknown	15.5	AD, FA (CM), AR	N.D.	5.8	2.1	5.2	5.4	Pass	Fail: anaphylaxis	N.D.
7	1	M	Anaphylaxis	Unknown	8.19	AD, Asthma	2.0	3.5	0.0	1.0	3.0	Pass	Excluded	N.D.
8	7	M	Urticaria	Unknown	4.98	AD, FA (wheat, walnuts, pine nuts), AR	N.D.	4.8	0.0	7.1	6.9	Pass	Fail: anaphylaxis	Excluded
9	2	F	Urticaria	Unknown	4.48	AD, FA (CM)	1.9	2.0	0.0	2.0	3.6	Pass	N.D.	N.D.
10	2	M	Urticaria	Unknown	4.32	AD	N.D.	9.0	2.5	5.0	5.5	N.D.	Excluded	N.D.
11	1	F	Urticaria	Unknown	3.80	AD, FA (CM)	2.0	1.2	0.0	0.0	1.2	Pass	Excluded	Pass
12	1	F	Urticaria	Unknown	1.77	FA (CM)	4.1	5.1	0.0	0.0	2.0	Pass	Excluded	Fail: rash, irritability

Abbreviations: SPT, skin prick test; OFC, oral food challenge; QEw, quail egg white; HEw, hen’s egg white; HEw-sIgE, hen’s egg white-specific IgE; AR, allergic rhinitis; AD, atopic dermatitis; FA, food allergy; CM, cow’s milk; PN, peanut; ND, not done. OFC exlu.

## Data Availability

Not applicable.

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
