# Peer review of "A Preliminary Study on Cross-Reactivity of Heat-Treated Quail and Hen’s Egg White Proteins in Young Children"

_nutrients, 2021, doi:10.3390/nu13072172_

Round 1

Reviewer 1 Report

The paper nutrients-1270959 is more a short communication than a research paper.

The study, which is preliminary, aimed to investigate the effects of different types of heat treatments on hen’s egg white (HEw) and quail egg white (QEw) proteins and their cross-reactivity in young children.

The title should be changed by adding “preliminary results”.

The Figures 1A and B are not relevant. Please remove or move as supplementary data.

The statistics of this paper are very weak.

The results of the gels 2 are not acceptable in the current form. The full gels should be used. Further, how can the authors be sure about the identity of the proteins they are mentioning without any mass spectrometry analysis?

The caption of Figure 3 needs much more work. Please detail it as much as possible. The same for any figure and table of this manuscript.

Reviewer 2 Report

This is a very useful study which aimed to increase our evidence based knowledge of alternative dietary choices for children with egg allergy.

The manuscript is well-written and I have minor amendment suggestions only.

Abstract:

The sentence “Domestic cooking barely altered the allergenicity” needs to be changed to ‘boiling egg had limited effect on altering the egg allergencity’. This is more specific as this study did not examine domestic cooking of egg in baked goods (for example cake) or in oven cooked dishes like quiche.

The last sentence needs to be modified as the inclusion of the word ‘safe’ maybe misinterpreted, especially given that 16.7% of the children still reacted to the stone-baked HEw and one child had anaphylaxis.

Introduction:

Line 52: Full stop is missing on line 52 after [7–9].

Line 65: Please replace “aimed to provide alternative dietary guidelines” with ‘aimed to provide alternative dietary choices’.

Results:

Lines 158-9: There is a typo for the IQR, 19.234.7 kU/L, which I think should be 19.2-34.7 kU/L?

Table 1: what does “Limit” mean? Please include a definition in the footnote of this table.

Line 182: Could “being heated” be clarified? Is this boiled? If so that that be changed to ‘being boiled’?

Lines 202-231: Many of these details are already clearly included in Table 1, thus could this results section be shortened to only include details that are not already in Table 1.

Discussion:

Lines 262-267: Please edit these sentences to remove the comments “improves the probability of achieving tolerance” and “that is beneficial for developing immune tolerance in HEw-anaphylaxis patients” as this was not investigated in this study, and has also not been shown overall in RCTs in other studies either.

Lines 268-278: Could these sentences please be moved to the methods section to give more details associated with line 84 “Stone-baked HEs were purchased from a local Korean market”.
